🔓 | **Open Peer Review** | Computational Biology | Research Article

# Cell behaviors underlying *Myxococcus xanthus* aggregate dispersal

Patrick Murphy,[1,2] Jessica Comstock,[3] Trosporsha Khan,[3] Jiangguo Zhang,[1,2] Roy Welch,[3] Oleg A. Igoshin[1,2,4,5]

**ABSTRACT** The soil bacterium *Myxococcus xanthus* is a model organism with a set of diverse behaviors. These behaviors include the starvation-induced multicellular development program, in which cells move collectively to assemble multicellular aggregates. After initial aggregates have formed, some will disperse, with smaller aggregates having a higher chance of dispersal. Initial aggregation is driven by two changes in cell behavior: cells slow down inside of aggregates and bias their motion by reversing direction less frequently when moving toward aggregates. However, the cell behaviors that drive dispersal are unknown. Here, we use fluorescent microscopy to quantify changes in cell behavior after initial aggregates have formed. We observe that after initial aggregate formation, cells adjust the bias in reversal timings by initiating reversals more rapidly when approaching unstable aggregates. Using agent-based modeling, we then show dispersal is predominantly generated by this change in bias, which is strong enough to overcome slowdown inside aggregates. Notably, the change in reversal bias is correlated with the nearest aggregate size, connecting cellular activity to previously observed correlations between aggregate size and fate. To determine if this connection is consistent across strains, we analyze a second *M. xanthus* strain with reduced levels of dispersal. We find that far fewer cells near smaller aggregates modified their bias. This implies that aggregate dispersal is under genetic control, providing a foundation for further investigations into the role it plays in the life cycle of *M. xanthus*.

**IMPORTANCE** Understanding the processes behind bacterial biofilm formation, maintenance, and dispersal is essential for addressing their effects on health and ecology. Within these multicellular communities, various cues can trigger differentiation into distinct cell types, allowing cells to adapt to their specific local environment. The soil bacterium *Myxococcus xanthus* forms biofilms in response to starvation, marked by cells aggregating into mounds. Some aggregates persist as spore-filled fruiting bodies, while others disperse after initial formation for unknown reasons. Here, we use a combination of cell tracking analysis and computational simulations to identify behaviors at the cellular level that contribute to aggregate dispersal. Our results suggest that cells in aggregates actively determine whether to disperse or persist and undergo a transition to sporulation based on a self-produced cue related to the aggregate size. Identifying these cues is an important step in understanding and potentially manipulating bacterial cell-fate decisions.

**KEYWORDS** bacterial development, biofilms, collective behavior, myxobacteria

Many bacterial species spend part of their life cycles as biofilms, surface-associated multicellular communities, which are resistant to harsh environmental conditions (1–3). Cells in these biofilms respond to environmental cues as well as biological signals produced by nearby cells to coordinate collective behaviors (4–6), change gene expression (7, 8), or undergo differentiation into distinct cell types (2, 9). Identifying

Address correspondence to Oleg A. Igoshin, igoshin@rice.edu.

Patrick Murphy and Jessica Comstock contributed equally to this article. Author order was determined by who drafted the initial paper.

The authors declare no conflict of interest.

See the funding table on p. 16.

how bacteria modulate their behavior in response to different cues can inform our understanding of the drivers of biofilm formation, restructuring, and stability. Biofilm formation by a Gram-negative bacterium *Myxococcus xanthus* is an important model system to understand these phenomena.

*M. xanthus* is a model organism for studying multicellular coordination due to its diverse range of emergent behaviors (10–12). These bacteria use both social motility (S-motility) and adventurous motility (A-motility) to move on surfaces. S-motility relies on type IV pili to attach to neighboring cells or extracellular polysaccharides (5, 13, 14), while A-motility uses membrane-bound focal adhesion sites to propel individual cells (13, 15–17). Cells periodically switch their motor's polarity, reversing their direction of motion, regardless of which motility system is used (18, 19). How cells adjust their motility systems is influenced by both contact-dependent signaling and chemoattractants, each of which transmits information about the nearby environment. Contact-dependent signaling conveys information about the local environment, such as cellular density and neighbor alignment, and also whether a cell is moving with or against its neighbors (6, 20–23). Chemotaxis in *M. xanthus* allows cells to climb gradients of external lipid concentrations, including phosphatidylethanolamine and diacylglycerol, by suppressing directional reversals (24–26). By using these signaling pathways to inform the use of their motility systems, *M. xanthus* cells can exhibit collective behaviors that include swarming (10), rippling (27–29), and multicellular development (30–32).

Under starvation conditions, *M. xanthus* cells undergo a multicellular development program, culminating in the formation of spore-filled fruiting bodies containing tens of thousands of spores (1, 30–33). This process occurs over roughly 24 h and involves distinct stages. Initially, cells exhibit low motility for several hours, after which, they display a burst of motion, coordinating into streams of cells via adjustments in speed and reversal frequency (33, 34). The intersections of these streams increase cell density and often results in initial aggregates appearing nearby. Over the next 5–7 h, cells build sequential layers that develop into the main mass of the nascent fruiting body. During this phase, several cell behaviors contribute to the growth of initial aggregates (35, 36). One of these behaviors is a density-dependent traffic jam effect, which slows down movement in high-density regions and increases the likelihood of cells entering a stopped state. Another important behavior is a biased random walk that influences cells in the vicinity of existing aggregates, with persistent cells moving for longer when oriented toward aggregates prior to reversing their polarity. These two behaviors interact synergistically to enhance aggregate growth. The steady influx of cells resulting from the biased random walk increases the local density, strengthening the jamming effect. Aggregate development culminates in the differentiation of a subset of the cellular population into environmentally resistant spores.

While the formation and protection of spores is the main goal of aggregation, not every initial aggregate is stable and some disperse prematurely (37–39). Cells abandon these unstable aggregates, migrating to other growing aggregates nearby. Thus, fruiting body development can be split into two phases: initial aggregation, where aggregates first appear and grow, and coarsening, where initial aggregates either disperse prematurely or remain and develop into full 3D structures with differentiated cell types (Fig. 1). The start of the coarsening phase varies but is typically 10–15 h after plating cells on agar (37–39).

It is unknown what triggers some aggregates to disperse midway through formation, but certain aggregate features are highly correlated to stability (38). Among these, size has been identified as the most distinctive feature that separates stable and unstable aggregates. As a result, predictions of aggregate stability using size have been reasonably successful (38, 39). However, no previous studies have quantified changes in cellular-level behaviors accompanying aggregate dispersal and their links to aggregate features.

In this study, we set to identify cell behaviors that drive aggregate dispersal. Based on our analysis of tracked cell data, we investigated how the traffic jam effect and the biased random walk vary as aggregates disperse. To identify the main source of dispersal, we

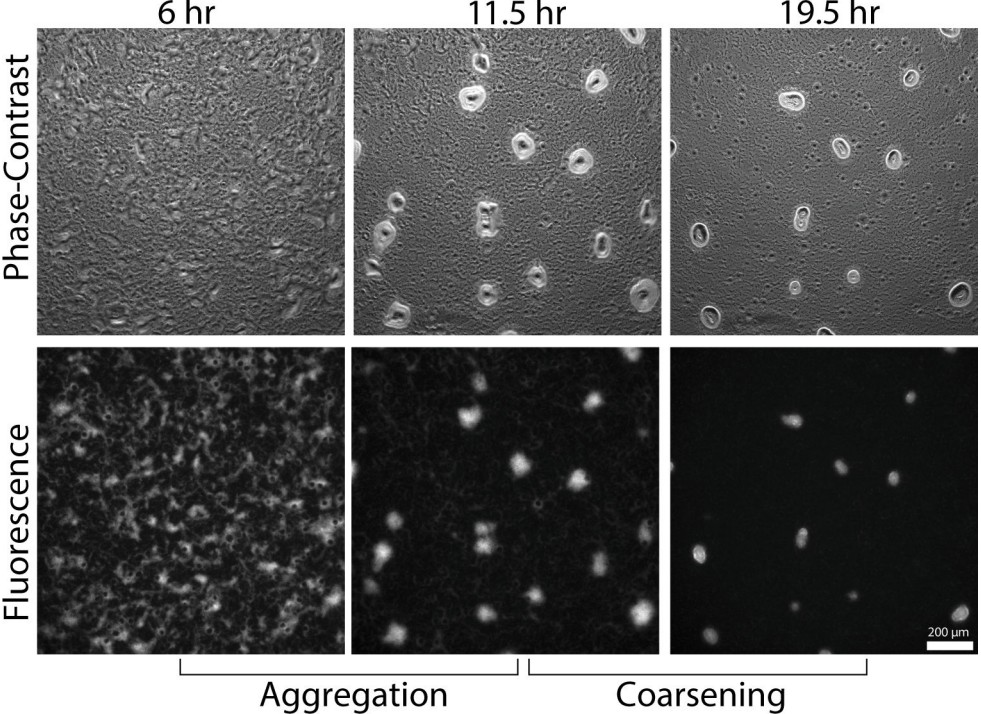

**FIG 1** Initial aggregation and aggregate dispersal over a period of 13.5 h starting 6 h after plating cells on starvation agar. (Top) Phase contrast images at 6, 11.5, and 19.5 h. (Bottom) Fluorescent images at the same time points. Initial aggregation starts around 6 h, continuing up to 11.5 h. At 11.5 h, some stationary aggregates begin to destabilize, continuing over 8 h until the last unstable aggregate has finished dispersing.

utilized a computational approach adapted from Cotter et al. (35). The results indicated that dispersal is primarily driven by the changes in the biased random walk—cells near smaller aggregates tend to move for longer when going away from the aggregates. Finally, to determine if the relationship between the biased random walk and aggregate dispersal is consistent across strains, we analyzed the behaviors of our tracked cells mixed with an alternate wild-type strain with low levels of dispersal and analyzed how cell behavior changed.

## RESULTS

### Quantification of aggregation and coarsening behaviors for the entire wild-type bacterial population

We employed fluorescent microscopy and a mixture of fluorescently labeled cells to collect data on cell trajectories and local cell density during both the initial aggregation and coarsening phases. Approximately 0.1% of cells were labeled with tdTomato (see Materials and Methods for more details). These cells were sparse enough to allow their trajectories to be extracted and used for behavior quantification. The remainder of the mixture consisted of green fluorescent protein (GFP)-expressing cells capable of producing many dispersing aggregates. These cells were used to detect aggregate locations and features, with GFP fluorescent intensity serving as a proxy for cell density.

To quantify tracked cell behavior, we adapted the approach taken in references (35) to coarse-grain cell trajectories. Each cell trajectory was divided into run vectors based on the cell's motile state: persistent or non-persistent. A cell was assigned a persistent state when moving steadily along its long axis, while a non-persistent state was assigned instead if the cell was determined to have little net displacement due to low velocity or high reversal frequency (see Materials and Methods for more details). Reversals in a persistent cell's direction of motion along its long axis were used to further divide

trajectories so that each run started and ended with either a change of state or a reversal in the persistent state. Cell behavior for each run was then quantified using the duration of the run, the cell's mean speed, the distance traveled, and the orientation. We also recorded data for a number of variables associated to each run such as time, local cellular density, local strength of cell alignment, the change in orientation between consecutive runs, cell position and orientation relative to the nearest aggregate, and the nearest aggregate features. Each of these variables represents a possible cue that *M. xanthus* cells could be using during aggregation. By investigating correlations between these cues and the cellular dynamics, we can infer what behaviors are important for aggregate development.

We first verified whether the cell behaviors identified in a previous study (35) were also present in our data since our mixture of fluorescent strains differed from theirs (see Methods and Materials for our strain details). The first check was to confirm that the mean duration of tracked cells' persistent runs was longer when cells moved toward aggregates compared to when they moved away. The second was to quantify the traffic jam effect, which manifests as a decreased run speed and an increased proportion of non-persistent cells inside aggregates. We started by quantifying the bias in the persistent random walk, termed the reversal bias, using the relative difference between the mean time spent moving toward the nearest aggregate, $t_{toward}$, and the mean time spent moving away, $t_{away}$

$$\text{reversal bias} = \frac{t_{toward} - t_{away}}{t_{all}}.$$

The value $t_{all}$ here is the mean duration of all persistent runs. Calculating the reversal bias in a 1-h moving window, we found that it was positive throughout initial aggregation (Fig. S1A, purple lines), indicating that cells bias their persistent movement toward aggregates as found previously (35, 36). We then quantified the strength of the traffic jam effect using the difference in persistent speeds, transition probabilities into the non-persistent state, and the durations of the non-persistent state between cells inside versus outside aggregates (Fig. S1B through D, purple lines). Measurements of all three metrics during initial aggregate development were similar to those reported previously (35, 36).

During the coarsening phase, our data analysis showed no significant changes in the behaviors mediating aggregate formation. The reversal bias dropped but remained positive, indicating cells were, on average, still biasing their motion toward aggregates (Fig. S1A, green lines). The measurements of the traffic jam effect showed slight changes, with the run speed increasing, the probability of entering the non-persistent state decreasing, and the duration of the non-persistent state rising slightly near the center of aggregates (Fig. S1B through D, green lines).

## Existing agent-based simulation without aggregate features captures aggregation but not dispersal

To test if the quantified cell behaviors could explain aggregate formation and dispersal, we used a data-driven agent-based model (ABM) (35). The approach follows the methodology of reference (35) and can be found in Materials and Methods. We ran ABM simulations using cell data drawn from a single experimental movie (see Movie S1). All simulations were run with the full set of cues for cell behavior used in reference 35: the current time, distance to the nearest aggregate, orientation relative to the nearest aggregate, local density, and the local strength of cellular alignment. In our simulation, agents formed initial aggregates over the course of 5.5 h. Aggregation continued over another 8 h, resulting in a set of final aggregates containing the majority of agent cells (Fig. 2A). The statistics of the six independent runs of the model were then averaged to capture their variation. The results indicate that during initial aggregate formation, the fraction of agent cells inside aggregates increased over time, matching the experimental fraction after a 3.5-h delay (Fig. 2B). The distribution of aggregate sizes in the simulation

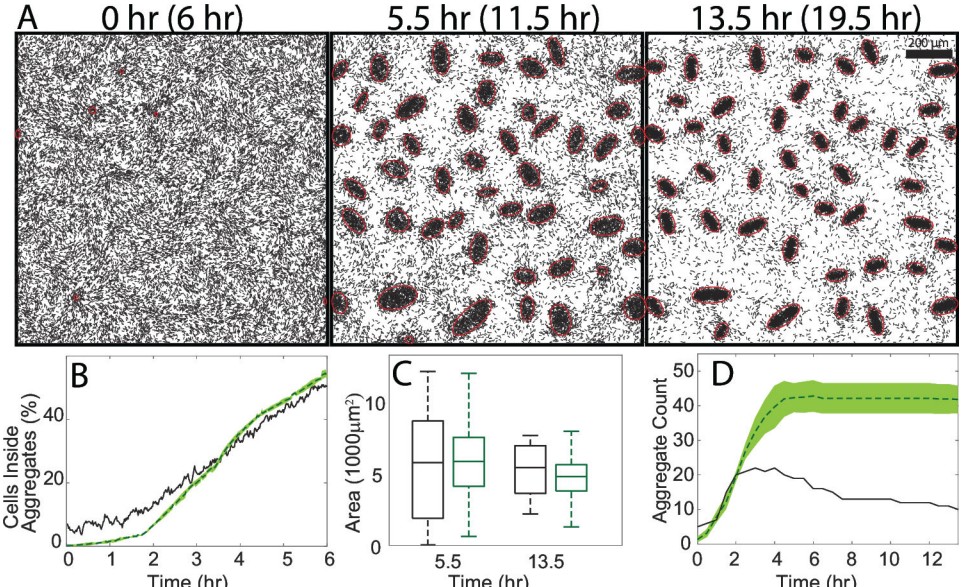

**FIG 2** (A) Simulation at the experimental equivalent of 6 h (aggregation start), 11.5 h (coarsening start), and 19.5 h (coarsening end) in the experiment. Detected aggregates are indicated in red. (B) Fraction of simulated cells in aggregates for the experiment (black) and simulation (green) during the initial aggregation phase. The shaded area for the green curve marks one standard deviation from the mean, which is quite small. (C) Aggregate area distribution for simulation (green) and experiment (black) at the start and end of the coarsening phase. (D) Aggregate count over the course of the simulation (green) and experimental equivalent (black). The shaded area marks the 95% confidence interval.

also agreed with the experimental distribution at the end of both the aggregation and coarsening phases, although the variance in size was less pronounced in simulations (Fig. 2C).

Despite good quantitative matches in the aggregation rate and sizes of aggregates produced, our simulations showed few aggregates dispersing (Fig. 2A, second and third panels). We confirmed that initial aggregation in simulations followed the experimental trend, although there were a greater number of aggregates that formed in simulations (Fig. 2D). This discrepancy was due to the appearance of new stable aggregates between hours 2 and 5 in the simulation. No new aggregates formed in the experiment past this time, with any increase in the aggregate count resulting from a motile aggregate settling down. Also, unlike the experimental movie where 50% of aggregates dispersed during the coarsening phase, the number of simulated aggregates remained unchanged after 5 h with little destabilization. Furthermore, the range of aggregate sizes produced in simulations was broad enough to include aggregate sizes that dispersed in the experiment. We conclude that correlations of the quantified cell behaviors with the local density, cell alignment, and distance to the nearest aggregate are insufficient to capture aggregate dispersal.

## Cell behaviors near stable and unstable aggregates during coarsening phase are distinct

To identify any key differences in cell behaviors around unstable aggregates, we separated our data set into two categories based on the nearest aggregate stability. We then systematically looked for differences in any of the cell behaviors known to be important for aggregation, such as those relevant to the traffic jam effect, the biased random walk relative to aggregates, and local cell alignment. This process identified two differences in cell behavior that could potentially destabilize an aggregate.

The first difference is that the reversal bias for cells near unstable aggregates is weaker in strength (Fig. 3A) and range of effect than near stable aggregates (Fig. 3B). This

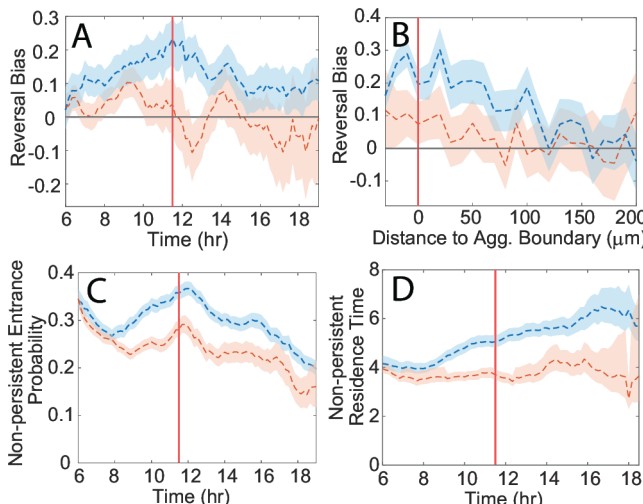

**FIG 3** (A) Reversal bias as a function of time for cells near stable (blue) and unstable (red) aggregates in the experiment. (B) Reversal bias as a function of the distance from the nearest aggregate boundary (set to be 0) for cells near stable (blue) and unstable (red) aggregates in the experiment. (C) Transition probability for entering the non-persistent state from the persistent state instead of reversing for cells near stable (blue) and unstable (red) aggregates. (D) Transition rate for exiting the non-persistent state for cells near stable (blue) and unstable (red) aggregates. The shaded regions denote 95% confidence intervals for the mean, and the vertical red lines mark the start of the coarsening phase.

strength difference also holds when only considering cells oriented less than 45 degrees away from the aggregate (Fig. S2A and B), indicating it is not just an effect from cells moving perpendicular to the aggregate. Additionally, around the start coarsening phase, the reversal bias decreases by approximately 0.1 near stable aggregates and by 0.2 near unstable aggregates (Fig. 3A). As the reversal bias for unstable aggregates is initially lower, this often leads to negative mean reversal bias values, indicating that cells move away from unstable aggregates for longer than toward them. This negative reversal bias is strong enough to deplete these aggregates of cells during the coarsening phase. We also observed that the decrease in the mean reversal bias was not maintained throughout the coarsening phase. The aggregates that caused the initial drop to negative bias disperse quickly, with the remaining unstable aggregates still having a positive mean reversal bias. However, when these remaining aggregates destabilize after 15 h, we observe a drop in the mean reversal bias again.

The second difference in cell behavior that we identified was a reduction in the proportion of cells in the non-persistent state around unstable aggregates, which was the result of two factors. First, cells near unstable aggregates showed a consistent decrease in the probability of transitioning to the non-persistent state instead of reversing (Fig. 3C). Second, these cells had a shorter residence time in the non-persistent state (Fig. 3D). Both the probability of stopping and the time in the non-persistent state increase with density, so these observations likely reflect that unstable aggregates are typically smaller and less dense than stable ones.

To incorporate stability-dependent cell behaviors in our simulations without prior knowledge of aggregate stability, we utilized the correlation between aggregate fate and size. Since smaller aggregates tend to be unstable and larger ones tend to be stable, we inferred that there were aggregate area-dependent effects on cells resulting in changes in reversal bias and the traffic jam effect (38). To test this, we plotted the reversal bias, transition probability to the non-persistent state, and non-persistent state residence time for the largest and smallest 50% of aggregates in the experiment (Fig. S2C through F). The results showed that separating the data by aggregate area and stability produced similar outcomes, indicating that the nearest aggregate size is a crucial cue for determining appropriate cell behaviors. The cells near small aggregates showed a

change in the sign of the mean reversal bias as aggregates destabilized, decreased the probability of entering a non-persistent state, and reduced residence times in that state.

The results of the above analysis suggested that our agent simulations did not properly account for some of the observed differences in cell behavior based on nearest aggregate size. We confirmed this for the reversal bias by comparing its strength in simulations near stable and unstable aggregates (Fig. S3A). The mean reversal bias in simulations matched the mean experimental reversal bias regardless of aggregate stability, indicating that the original agent-based model lacked the necessary cues to properly select cell behavior. Although our model used cues such as local density and distance from the nearest aggregate boundary that could theoretically capture some size-dependent effects, these cues were insufficient to replicate aggregate dispersal.

## Aggregate dispersal is recovered by the inclusion of the nearest aggregate area in the simulation's cues for cell behavior

To investigate the effects of size-dependent variations in the reversal bias and the traffic jam effect on aggregate stability, we incorporated the nearest aggregate area as a cue for determining the state and state duration of agent cells in our simulations. This inclusion resulted in aggregate dispersal during the equivalent of the experimental coarsening phase (Fig. 4A and B). The observed decrease in simulated aggregate number during the coarsening phase was on average about 30%–40%, close to the observed value of 50% seen in the experiment. Furthermore, aggregate sizes at the start of the coarsening phase still matched well with experiments (Fig. 4D). Finally, creating a logistic model of aggregate stability using size as the explanatory variable revealed that unstable aggregates produced by our updated ABM were typically small (Fig. 4D), in line with the experimental findings.

Although the updated ABM produced dispersal, it did not perfectly match all experimental observations. Notably, our simulations still produced more aggregates than seen experimentally (Fig. 4B). Also, the size of aggregates by the end of the

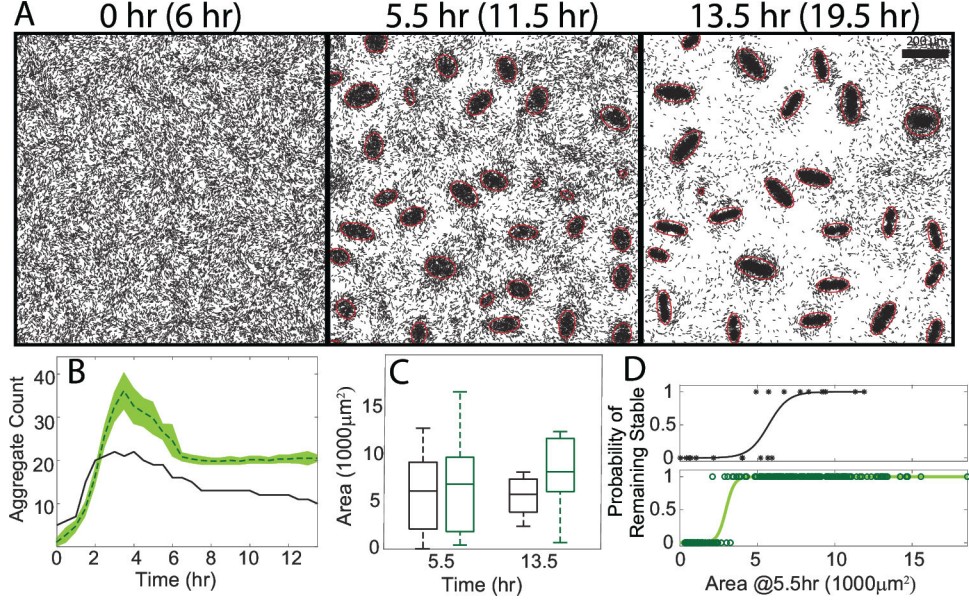

**FIG 4** (A) Simulation at the experimental equivalent of 6 h (aggregation start), 11.5 h (coarsening start), and 19.5 h (coarsening end). Detected aggregates are indicated in red. (B) Aggregate count over the course of the simulation (green) and experimental equivalent (black). The shaded area marks the 95% confidence interval. (C) Aggregate area distribution for simulations (green) and experiment (black) at the start and end of the coarsening phase. (D) Logistic regression using aggregate fate (stable or unstable) versus aggregate area at the start of the coarsening phase for the experiment (black) and the combined simulations (green). Asterisks mark the data points for the experiment, and circles mark the data points for the simulations.

simulation was greater than those in the area-independent simulations and no longer matched the experimental data (Fig. 4C). Since larger aggregates possess a stronger reversal bias and can grow faster, having slightly larger aggregates in simulations early on can snowball into having a moderate size discrepancy by the end. There were two minor additional discrepancies regarding aggregate dispersal. Our updated simulations produced unstable aggregates during the coarsening phase that were typically smaller than those seen in the experiment, and some mid-sized dispersing aggregates were still present by the end of the simulations. Since aggregate area alone does not cleanly separate stable and unstable aggregates in the experiment, it is likely that our simulations mixed together cell behaviors near both stable and unstable mid-sized aggregates, producing mid-sized unstable aggregates with longer dispersal times.

To identify whether the area-dependent reversal bias or the area-dependent traffic jam effect played a greater role in dispersal, we performed two sets of simulations with each behavior isolated. The first set included only an area-dependent reversal bias as a cue for an agent's run speed and duration, while the second set included only an area-dependent traffic jam effect as a cue for an agent's next motile state. We found that including only an area-dependent reversal bias accounted for nearly all dispersal seen previously. The agent cells also displayed a mean reversal bias that qualitatively matched the experimental data (Fig. S3B). The quantitative discrepancy in the reversal bias was likely due to differences in the distribution of simulated aggregate sizes, which now directly affects their development. However, running the simulations with just the area-dependent jamming resulted in very low levels of dispersal equivalent to those seen in the area-independent simulations (Fig. 5A) despite maintaining a wide range of developing aggregate sizes (Fig. 5B). Lastly, we found that the area-dependent reversal bias maintained the same size threshold of 3,000 $\mu m^2$ for stable aggregates (corresponding to a value of 0.5 on the logistic curve), while the area-dependent jamming decreased that threshold significantly (Fig. 5C). We conclude that the observed differences in cells' non-persistent states based on aggregate size do little to affect aggregate dispersal, while the dependence of the mean reversal bias on size accounts for the majority of dispersal.

One discrepancy that persisted after simplifying the ABM to include an area-based reversal bias was that fewer mid-size aggregates dispersed compared to experiments. Many aggregates this size shrunk during the 8 h coarsening phase of our simulations, but they often did not fully disperse by the end. By extending the end of the simulated coarsening phase by 7.5 h to match 27 h experimentally, we were able to fully destabilize more mid-sized aggregates, increasing the mean stable size threshold to 4,000 $\mu m^2$, closer to the experimental threshold of 5,000 $\mu m^2$ (Fig. 5C).

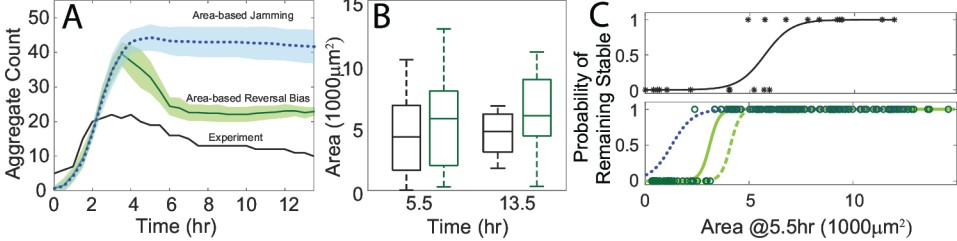

FIG 5 (A) Aggregate count over the course of the area-based reversal bias simulation (green), area-based jamming simulation (blue), and the experimental equivalent (black). The shaded areas marks the 95% confidence intervals. (B) Aggregate area distribution for the area-based reversal bias simulation (green) and experiment (black) at the start and end of the coarsening phase. (C) Logistic regression using aggregate fate (stable or unstable) versus aggregate area at the start of the coarsening phase for the experiment (black), the combined area-based reversal bias simulations (solid green), the extended area-based reversal bias simulations (dashed green), and the combined area-based jamming simulations (dotted blue). Asterisks mark the data points for the experiment, and circles mark the data points for the non-extended area-based reversal bias simulations (green).

## Dispersal mechanism is the same across experimental replicates despite heterogeneity in aggregate dispersal, timing, and sizes of unstable aggregates

Our cell mixture produces aggregates that can vary wildly in size from replicate to replicate. Because of this, combining data sets with different aggregate sizes can affect the unstable aggregate size threshold and initial aggregate sizes, resulting in diminished dispersal even with the updated model. We designed our initial analysis around the behaviors of a specific replicate (data set 1) to avoid averaging size-dependent behaviors and potentially obscuring their source (Fig. S4A and B). Therefore, to expand our results, we repeated our data analysis and simulations individually on time-lapse data from two additional experiments (data sets 2 and 3).

The aggregate-level analyses of the replicates revealed that the mean size of unstable aggregates ranged from less than 1,000 $\mu m^2$ to over 3,000 $\mu m^2$ (Fig. 6A). Interestingly, the mean size of unstable aggregates was smaller when the population consisted of smaller aggregates, and larger for populations of larger aggregates. This correlation suggests that aggregate dispersal depends on the relative size of aggregates rather than on a fixed size threshold that is consistent across experiments.

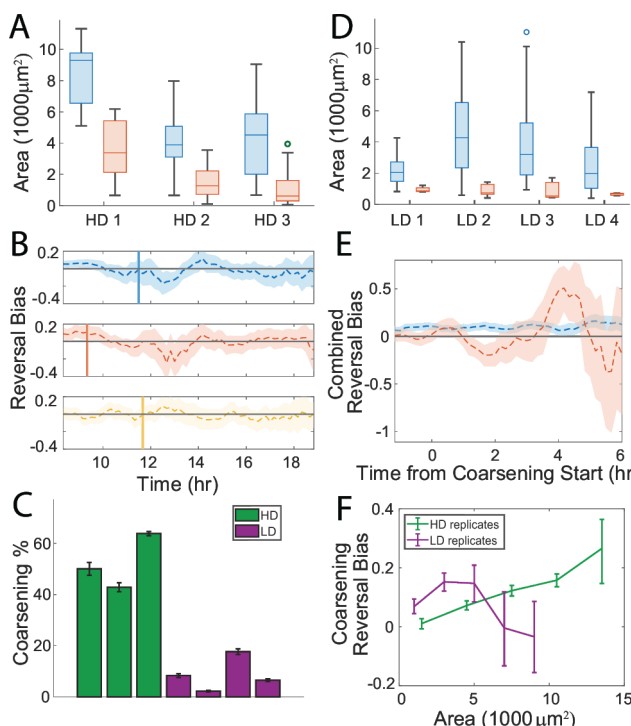

**FIG 6** (A) Distribution of stable (blue) and unstable (red) aggregate areas at the start of the coarsening phase for the high dispersal (HD) replicates. The mean sizes of both populations scale together. (B) The reversal bias for cells near the aggregates below a threshold size for the HD strain replicates: set 1 (red), set 2 (blue), and set 3 (yellow). All three means decrease near the start of the coarsening phase, increase as the first wave of unstable aggregates disperse, then drop again as the rest of them start to destabilize. Size thresholds (smallest 50%, 30%, and 50%, for sets 1, 2 and 3 respectively) were chosen to capture this first wave of dispersing aggregates. (C) Percentage of dispersing aggregates for the HD strain (green) and the low dispersal (LD) strain (purple). (D) Distribution of stable (blue) and unstable (red) aggregate areas at the start of the coarsening phase for the LD replicates. (E) Reversal bias for tracked cells near stable (blue) and unstable (red) LD strain aggregates combined across all four replicates and aligned at the start of the coarsening phase (0 h on the horizontal axis). (F) Reversal bias during the coarsening phase varies with aggregate size for HD and LD aggregates (data from the all replicates combined). The reversal bias for small aggregate in the LD mixture is highers than the HD mixture.

The experimental replicates differed in both the mean aggregate size of dispersing aggregates and the start of the coarsening phase, leading us to conjecture that the reversal bias could have also changed in strength or timing. We investigated the biased random walks near smaller aggregates, finding that the mean reversal bias dropped in a window of about an hour from the start of the coarsening phase despite differences in the size of aggregates and the timing of dispersal (Fig. 6B). As time progressed, the reversal bias again fluctuated between negative and positive values based on the number of unstable aggregates currently dispersing. The total percentage of the aggregates that dispersed among the three data sets was consistently high, ranging from 40% to 60% (Fig. 6C green bars). These results indicate that changes in the reversal bias serve as a consistent mechanism for dispersal and suggest that our ABM, using tracking data from each experiment, should also reproduce aggregate dispersal in that experiment.

To test if the reversal bias was the driving factor behind aggregate dispersal across experiments, we ran our agent-based simulations using both additional data sets. We found that simulations implementing area-based reversal bias faithfully produced good approximations for both the percentage of unstable aggregates and their size threshold regardless of the data set used. These simulations showed both high levels of aggregate dispersal (Fig. S5A) and a high size threshold for stable aggregates (Fig. S5B). Interestingly, running simulations with no dependence on area produced dispersal levels of ~20% (data set 2) and ~60% (data set 3), with the likeliest cause of data set 3's high dispersal (HD) percentage being the high proportion of cell data near unstable aggregates. Not unexpectedly, with few exceptions, the actual number of aggregates produced still had some inaccuracies when compared to experiments, especially when the area was used in the persistent state duration search (Fig. S6 and S7).

## Analysis of a second wild-type strain implies genetic changes that affect the reversal bias change aggregate dispersal

It was previously demonstrated that interlaboratory evolution of DK1622 strains has led to some phenotypic differences between wild-type strains (40). To investigate whether other wild-type strains exhibit different coarsening phenotypes, we conducted a study comparing various wild-type strains with ours. We found a low dispersal (LD) strain that consistently showed a reduced level of dispersal compared to our HD strain (Fig. S8; Movie S2), providing an opportunity to compare how tracked cell reversal bias differs in different mixtures. Over four experimental replicates, the LD strain displayed an average dispersal level of less than 10% (Fig. 6C, purple bars), producing a higher proportion of small, stable aggregates compared to the HD strain. This finding suggests that genetic differences between the two strains affect aggregate dispersal and could shift tracked cell behaviors related to aggregate destabilization.

To determine if tracked cells change their reversal bias during dispersal in a consistent way in both the HD and LD strains, we analyzed cell behavior in the LD mixture using the same methodology as before. Due to the small number of unstable aggregates and cells per aggregate in each LD data set, the mean reversal bias near unstable aggregates was not statistically significant from zero. To circumvent this, we combined the cell data among LD replicates, aligning them in time based on the start of the coarsening phase. Our findings show that the reversal bias in the LD mixture (Fig. 6E) and the HD mixture (Fig. 5A) match qualitatively. During the coarsening phase, tracked cells in the LD mixture, on average, exhibit a biased random walk toward stable aggregates and away from unstable aggregates.

Since LD aggregates rarely destabilize even if they are small, it is not apparent that the mechanism for dispersal is the same. A time-dependent change in the reversal bias was previously required for the destabilization of aggregates, so we hypothesized that if our tracked cells respond to the LD mixture consistently, then their reversal bias near small aggregates must remain positive on average. To test this hypothesis, we compared the reversal bias of tracked cells near aggregates of similar sizes between both mixtures

during the coarsening phase. We found that cells near small LD aggregates exhibit a higher mean reversal bias during coarsening than near small HD aggregates (Fig. 6F). As the aggregate size is increased, the mean reversal bias of cells near HD aggregates increases steadily, overtaking the bias of cells in the LD mixture. Overall, this observation suggests that the level of aggregate dispersal in different strains is controlled by aggregate size-dependent variations in the strength of the reversal bias.

## DISCUSSION

In this study, we showed that the premature dispersal of *M. xanthus* aggregates is driven by cells changing their reversal bias to move away from aggregates. Our analysis was done in several stages, the first of which focused on investigating cell trajectories in a high dispersal mixture. We quantified how cell behaviors changed both between the vicinities of stable and unstable aggregates and between the initial aggregation and coarsening phases. This resulted in a short list of relevant behaviors, out of which, only differences in the reversal bias proved relevant to dispersal. The maximal drop in the mean reversal bias was about 0.2 (Fig. 3A), corresponding to roughly a 20% drop in the average reversal period when moving toward aggregates. This might seem like a small change; however, previous studies have shown that despite high levels of heterogeneity in the behavior of individual cells, such small changes in mean cell behavior can dramatically affect aggregation (35). This was especially true regarding the effect of a bias toward aggregates on the aggregation rate, so, unsurprisingly, a small bias away from aggregates can disrupt aggregate formation to the extent seen. Importantly, the strength of the bias away was enough to overcome the drop in motility inside aggregates. Since aggregation is theoretically possible, if more gradual, with just a density-driven slowdown (41), any dispersal mechanism must either circumvent or overcome this slowdown.

To measure the impact of cell behaviors on aggregate dispersal, we implemented a series of agent-based simulations focused on gauging whether different behaviors contributed to dispersal. We found that dispersal is essentially unaffected by the observed changes in the non-persistent state for cells in small aggregates (Fig. 5). However, it is worth noting that the proportion of non-persistent cells in small aggregates was still elevated compared to outside aggregates. Similarly, a previous study showed that longer, non-persistent stops inside aggregates are not the main reasons for successful aggregation. These two results together suggest that decreasing the proportion of non-persistent cells in aggregates has little impact on the initial or coarsening phases of aggregate development, with the reversal bias playing a more influential role in both phases.

The genetic basis of the reversal bias observed in tracked cell behavior near *M. xanthus* aggregates is unknown, but the bias is reliably demonstrable and differs significantly when tracked cells are mixed with HD or LD strains. We hypothesize that the reversal bias is the result of an unidentified attractant signal originating from the cells within each aggregate, which results in a self-reinforcing cycle of aggregation by increasing the bias toward each aggregate as the number of cells composing it increases. Evidence supporting this hypothesis is based on the observation that the distance from the aggregate boundary to where the reversal bias vanishes increases with an aggregate size (Fig. S2B), indicating a direct relationship between the proposed signal and the number of cells in the aggregate. The dependence of bias strength on aggregate size (Fig. S2A) further suggests that the corresponding signaling pathway may not be capable of robust adaptation (42–44). However, the average size of unstable aggregates is based on their relative size to the average size of all aggregates. Therefore, the signaling pathway would need to be capable of population-level sensing since the size threshold for unstable aggregates scales with the size of all aggregates (Fig. 6A). Therefore, the reversal bias would result from local differences from a baseline signal across the whole population. There is evidence from prior work in support of *M. xanthus* exhibiting chemotaxis, both from experiments (24–26) and computational studies (45),

but full molecular pathways have not been determined. Regardless of what the signal and the molecular pathway are, our results indicate the sensing mechanism must be capable of measuring both an aggregate's absolute and relative size.

By analyzing the behaviors of tracked cells in the LD mixture, we showed that the reduction in dispersal is linked to an increase in reversal bias toward aggregates (Fig. 6F). This link indicates that the dependence of the reversal bias on aggregate size is genotype dependent, but on its own does not identify a gene or set of genes that control aggregate dispersal. It is possible that one or more of the genes that differ between the LD wild type and HD wild type directly control aggregate dispersal. However, it is also possible that the reduction in dispersal results from compensatory changes in gene expression in response to mutations that don't directly affect the reversal bias. Further studies analyzing the differences in gene expression between the two strains could help resolve this uncertainty. In the future studies, this approach to identifying the biological source of the reversal bias could be supplemented by additional mutant screening, fractionating cells to test the response of aggregates to different molecular subsets, and testing how aggregation dynamics vary in different chemical and physical environments using, for instance, a flow chamber.

If genetically controlled signaling is responsible for the observed reversal bias, the natural question is what triggers the change in bias that leads to aggregate dispersal. We speculate that *M. xanthus* might change either its production of or its response to a hypothetical signal due to some time-dependent change in gene expression (8). The near-simultaneous drop in the bias strength regardless of aggregate size and stability (Fig. 3A and Fig. S2A) could then result from cellular differentiation or a loss of cells due to programmed cell death during aggregate development (46, 47). The evidence from our time course data is that the drop happens at a similar time to both the development of aggregate layers and the start of cell lysis. A change in gene expression, and thus response, is more likely than a change in production since the gradient produced by a chemotactic signal is unlikely to change significantly on the timescale the reversal bias drops.

While changes in gene expression provide a reasonable hypothesis for what triggers dispersal, it does not explicitly explain why cells actively migrate away from dispersing aggregates rather than gradually reverting to a bias-free random walk. If cells abandoning a dispersing aggregate changed their response to move away from a signal source, then it seems reasonable that they would actively avoid approaching another neighboring aggregate. However, abandoning cells are observed to move toward neighboring aggregates (39). This observation could be accounted for if small aggregates stop producing a strong signal and there are also nearby aggregates capable of influencing nearby cells. The caveat to this conjecture is that some aggregates disperse with no neighboring aggregates nearby, suggesting that whatever factors cells use to determine if they should disperse are internal to the aggregate. This is supported by a previous study on aggregate stability that showed that proximity and size of neighboring aggregates have minimal effect on stability (38).

As an alternative to the chemotaxis hypothesis, an aggregation model based on Ostwald ripening (48, 49) was previously developed as a feasibility test for the minimum number of genetic inputs required to achieve both initial aggregation and dispersal (39). It succeeded in predicting the appearance and relative stability of developing *M. xanthus* aggregates with high accuracy, but the requisite conditions for Ostwald ripening do not match observations of cell motility and the active nature of developing cells in general. These discrepancies accumulate over time so that predictions based on the Ostwald ripening model tend to deviate significantly from predicted aggregation behavior over longer timescales (50).

Studying emergent behavior in microbial communities involves analyzing differences in cell behavior that result in significant shifts in cell dynamics. These changes are influenced by various factors including mechanical forces, signaling pathways, and alterations in gene expression. Our findings suggest that *M. xanthus* aggregation is

sensitive enough that a modest difference in how reversal bias changes near small and large aggregates can lead to a large difference in dispersal levels between wild-type strains. There may be no specific genetic mechanism directly controlling this change in reversal bias, meaning that any number of otherwise inconsequential genetic differences could cause a strain to have high or low dispersal rates. Those differences would only become important if evolutionary selection favored one set of differences over another. For *M. xanthus* bacteria, laboratory strains are typically discarded only if they fail to produce aggregates for no discernable reason, removing selective pressure from strains that undergo coarsening but still produce final aggregates. These findings may have implications beyond *M. xanthus* for other developmental model organisms displaying emergent behaviors.

## MATERIALS AND METHODS

### Strains and culture conditions

Three different *M. xanthus* DK1622 strains were used in this study. Both tdTomato-expressing LS3908 (35) and GFP-expressing DK10547 (27) commonly exhibit ~50% of aggregates dispersing during the coarsening phase, while S4 (40) shows minimal aggregate dispersal. We refer to DK10547 as the high dispersal wild-type strain and S4 is referred to as the low dispersal wild-type strain. tdTomato-expressing LS3908 was diluted into both the high and low dispersal strains to track differences in cell behavior.

All cells were grown overnight in CTTYE broth [1% casein peptone (Remel, San Diego, CA, USA), 0.5% Bacto yeast extract (BD Biosciences, Franklin Lakes, NJ, USA), 10 mM Tris (pH 8.0), 1 mM $KH(H_2)PO_4$, 8 mM $MgSO_4$] at 32°C with vigorous shaking. tdTomato-expressing strain LS3908 and GFP-expressing strain DK10547 were supplemented with 10 µg/mL oxytetracycline and 40 µg/mL kanamycin, respectively, for selection. Additionally, LS3908 was supplemented with 1 mM isopropyl β-D-1-thiogalactopyranoside (IPTG) to induce tdTomato expression.

To identify cell behaviors linked with aggregate dispersal, we set up development assays with a fraction of LS3908 cells diluted into the high dispersal strain DK10547. LS3908 cells were used for tracking individual cell behaviors while aggregate position and local density estimations came from the DK10547 cells. Cells were harvested from overnight CTTYE cultures as described above at mid-log phase, washed twice in TPM starvation buffer [10 mM Tris (pH 7.6), 1 mM $KH(H_2)PO_4$, 8 mM $MgSO_4$], and resuspended in TPM buffer to a cell concentration of $5 \times 10^9$ cells/mL. LS3908 cells were diluted 1:800 into high dispersal strain DK10547. A 5 µL droplet of cells was spotted on agarose slide complexes as previously described (51), containing 1% agarose-TPM medium supplemented with 1 mM IPTG. Imaging conditions are described below.

Low dispersal strain S4 reproducibly displays minimal aggregate dispersal during the coarsening phase. LS3908 was diluted 1:800 in the low dispersal S4 strain to determine if LS3908 cells changed their behavior when placed in a low-dispersing population. Development assays were performed on the LS3908-S4 mixture on agarose slide complexes as described above. Since the low dispersal S4 strain does not have a fluorescent label, local cell density estimates and aggregate segmentation were done using autofluorescence in the GFP channel.

### Time-lapse capture

The data presented in this paper represent three replicates of LS3908 mixed with DK10547 and four replicates of LS3908 mixed with S4, collected under the same experimental conditions on different days. Imaging was performed on a Nikon Eclipse E400 microscope with a pco.panda 4.2 sCMOS camera and NIS-Elements software. For cell tracking experiments, LS3908 samples were imaged with 400 ms exposure with a Sola LED light source at 75% intensity, and DK10547 and S4 samples were imaged with 200 ms exposure at 35% intensity. Control of the fluorescent filter wheel and

autofocus mechanism was managed with a MAC6000 system filter wheel controller and focus control module (Ludl Electronic Products, Ltd.). Images in the phase contrast and tdTomato channels were captured every 60 s over 24 h. Since aggregates do not change significantly on a timescale of minutes and frequent blue-light exposure can slow aggregation dynamics, images in the GFP channel were captured every 15 min to track the position of aggregates and changes in local cell densities.

The three replicates of the LS3908-DK10547 mixture were selected from a larger set of experiments based on two criteria: (i) the experiment must show coarsening (this phenotype occurs about 50% of the time for DK10547) with the time-lapse images capturing the whole coarsening phase (sometimes dispersal continued past the end of the movie), and (ii) there must be enough tracked cells during the coarsening phase to run simulations (simulations usually require several thousand cell runs every 2–3 h of simulated time). The second criterion was established since tracked cells disappear over time due to cell lysis, diminished fluorescence, or were being hidden inside dense aggregates. The first criteria removed about half of our initial 24 replicates from consideration, while the second narrowed it further to three replicates. For the movies of the LS3908-S4 mixtures, these two criteria were not applicable. Therefore, no selection was performed to acquire those four replicates.

## Aggregate detection and thresholding

The mean intensity of each image was subtracted, then the total intensity of the image was scaled to 1 so that the total intensity from frame to frame remained invariant. For consistency, a single time point was chosen to determine a threshold for segmenting aggregates from the background density. The chosen time point was halfway between aggregate initiation and the start of the coarsening phase. The corresponding image was rescaled so its maximum intensity was 1 and its minimum was 0; after which, Otsu's method (51) was used to calculate the threshold $I_{thresh}$. This threshold was then rescaled using the minimum and maximum pixel values, $a_{min}$ and $a_{max}$, the image had when its total intensity was 1. This produced a threshold $I_{scaledthresh} = (a_{max} - a_{min})*I_{thresh} + a_{min}$ that could be used on every frame in the movie. Aggregates were then segmented and tracked following reference (35). We then removed both aggregates that were only partially in the field of view and cells near those aggregates from the data set to prevent the data sets from containing inaccurate correlations between cell behaviors and aggregate size.

## Identification of the start of the coarsening phase

The start of the coarsening phase was identified by first filtering out motile and short-lived aggregates during the start of the initial aggregation. Motile aggregates were defined as those whose centroid moved more than 3 µm per minute, while short-lived aggregates were identified as those that either were stationary and dispersed after less than 2 h from initial detection or dispersed while counted as motile aggregates. Further filtering out stable aggregates left a set of unstable, non-motile, long-lived aggregates in each frame. The start of the coarsening phase was then defined as when the total area of these unstable aggregates started to continually decrease. This was typically between 9 and 12 h after the initial plating of cells on starvation agar.

## Cell behavior data extraction

To quantify cell behavior, we performed the same procedures found in reference 35 to track fluorescently labeled tdTomato cells and classify both cellular transitions between non-persistent and persistent states and reversals in the persistent state. Each trajectory segment with a start and end defined by a state transition or reversal, called a cell run, was then labeled, and the cell position, orientation, speed, and local alignment to other cells were recorded in a database. We only collected cell behaviors after initial aggregate formation at 4–5 h after plating. We then augmented the cell run database with the

nearest aggregate size, position, distance, and relative orientation from the cell at the start and end of each run.

## Agent-based model implementation

We implemented an extension of the agent-based model found in reference 35, which used a simulated domain equal in size to the experimental field of view. This model assumes that cells move in straight lines between reversals and stops, with a change in orientation upon each state transition. Therefore, an agent cell's behavior was determined by its change in motile state, its change in orientation when switching states, and the speed and duration of its next motile state.

The implementation of the closed-loop ABM relied on sets of internally measured cues to determine agent cell behavior. These cues were time, local density, cell state (persistent or non-persistent), the nearest aggregate area, distance from the nearest aggregate boundary, the relative angle to the nearest aggregate, and strength of local cell alignment. The strength of local alignment ($\gamma$) was calculated for each cell by first finding all neighboring cells in a 12 µm radius in space and a 7-min window backward in time. We then calculated a mean nematic angle $\bar{\theta}$ following reference 35. Finally, we calculated $\gamma$ as

$$\gamma = \cos\big(2(\theta_i - \bar{\theta})\big),$$

where $\theta_i$ is the orientation of the $i$th tracked cell.

The cues listed above were used as search variables in nearest neighbor (NN) searches of the compiled database of experimental cell runs. Three sequential searches were performed to determine an agent cell's state transition, change in orientation, and both the speed and duration of its next run. We used multiple searches since changes in orientation, speed, and duration depend on the agent's state, and since speed and duration further depend on an agent's relative orientation to aggregates after reorienting. Using the methods in reference 35 as guidelines, we implemented cues in the same way in our simulations with the exception of time and local density. The necessary timespan needed to simulate both aggregation and aggregate dispersal was, on average, about 13–15 h, nearly three times longer than initial aggregation alone. Since each cue was weighted equally when performing NN searches, longer simulations resulted in time differences receiving less weight, producing agent behavior that was out of sync. To avoid this, we binned cell behavior in the database into intervals and performed the NN search within each bin. Pre-coarsening phase time intervals were 100 min in length due to the more rapid changes occurring during initial development, while the coarsening phase was divided into two bins of equal length due to more gradual changes in time and fewer cell data in total. This binning approach had the additional advantage of speeding up the simulation by reducing the scopes of the NN searches. Local density was implemented during initial aggregation as in reference 35, but it was removed as a cue during the coarsening phase to reduce search parameters and speed up the simulation. As aggregates have already formed by that point, the distance from the aggregate boundary captures the local density accurately enough to search correctly for cell behaviors. Depending on the type of simulation, the cues used in the three NN searches were supplemented as needed with the nearest aggregate area.

Each simulation started with randomly placed cells in the simulated field of view at the equivalent of 6 h in the movie. We then ran a 90-min initialization using only two parameters: local density and local cell alignment. This initialization helps develop initial variations in density and gives cells time to align into streams. Cell behaviors in the initialization were sampled from the first 20 min of the experimental data. The agent cell positions and orientations at the end of the initialization were then used as initial conditions for the main simulation.

## Selection of bandwidth and aggregate size threshold in simulations

The density profile of agent cells in simulations was calculated using a kernel density estimator with a fixed bandwidth. We choose this bandwidth so that (i) the fraction of simulated cells inside aggregates matched the experimental equivalent by the start of the coarsening phase and (ii) the size of aggregates at the beginning of the coarsening phase approximated that seen in the equivalent movie. We selected a bandwidth of 7 µm for HD data set 1 and a smaller bandwidth of 5 µm for the replicate HD data sets 2 and 3. This smaller bandwidth for the second and third data sets was needed to reproduce the numerous small aggregates seen in those experiments.

The aggregate threshold inside simulations was selected similarly, starting with the threshold used to segment aggregates in the movies and adjusting slightly as needed. For HD data set 1, the threshold remained unadjusted from 2.16e–6. For HD data sets 2 and 3, which had a bandwidth of 5 µm, the density threshold was raised to 2.32e–6 to help control the initial number of aggregates from becoming too high. We explored using a greater number of agent cells as an alternative to changing the threshold but ultimately found it inferior due to it causing a discrepancy in measured local cell alignment between the simulations and the experimental data. This discrepancy resulted from having a much denser set of cells with which to measure the local alignment in simulations.

Density fluctuations in simulations sometimes resulted in the formation of pinpoint aggregates with unrealistically small sizes. Since cell behavior is determined by the nearest aggregate, such aggregates are undesirable. We filtered out these pinpoint aggregates by removing any detected aggregate below 300 $\mu m^2$ when calculating the aggregate nearest a given cell. This threshold was confirmed to remove noisy artifacts while still being small enough to initiate aggregate formation.

### ACKNOWLEDGMENTS

This research was primarily funded by NSF DMS-1903275 and IOS-1856742 (to O.I.), and DMS-1903160 and IOS-1856665 (to R.W.) awards with partial support of NSF IOS-1951025.

J.C. and T.K. performed the experiments. P.M. and J.Z. performed the data analysis and simulations. P.M., J.C., T.K., J.Z., R.W., and O.I. wrote the manuscript.

### AUTHOR AFFILIATIONS

[1]Department of Bioengineering, Rice University, Houston, Texas, USA
[2]Center for Theoretical Physical Biology, Rice University, Houston, Texas, USA
[3]Department of Biology, Syracuse University, Syracuse, New York, USA
[4]Department of Chemistry, Rice University, Houston, Texas, USA
[5]Department of Biosciences, Rice University, Houston, Texas, USA

### AUTHOR ORCIDs

Patrick Murphy ⓘ http://orcid.org/0000-0003-1321-8749
Jessica Comstock ⓘ http://orcid.org/0009-0003-7250-6887
Roy Welch ⓘ http://orcid.org/0000-0002-9946-108X
Oleg A. Igoshin ⓘ http://orcid.org/0000-0002-1449-4772

### FUNDING

| Funder | Grant(s) | Author(s) |
|---|---|---|
| National Science Foundation (NSF) | 1856742, 1856665, 1903275, 1951025, 1903160 | Patrick Murphy |
| | | Jessica Comstock |
| | | Trosporsha Khan |
| | | Jiangguo Zhang |

| Funder | Grant(s) | Author(s) |
|--------|----------|-----------|
|        |          | Roy Welch |

## AUTHOR CONTRIBUTIONS

Patrick Murphy, Formal analysis, Investigation, Methodology, Writing – original draft | Jessica Comstock, Data curation, Investigation, Writing – review and editing | Trosporsha Khan, Investigation, Writing – review and editing | Jiangguo Zhang, Investigation | Roy Welch, Conceptualization, Supervision, Writing – review and editing | Oleg A. Igoshin, Conceptualization, Supervision, Writing – review and editing

## DATA AVAILABILITY

All simulation and visualization codes are written in MATLAB. Both the processed data and the code used to analyze the data and run simulations are available online at Zenodo. The TIFF images recorded in experiments and used in this study are available on Zenodo. The tdTomato images used for cell tracking and the GFP images used for aggregate segmentation and density estimation of the high-dispersal cell mixture can be found at https://doi.org/10.5281/zenodo.8166434, while the fluorescent images for aggregate identification and tracking in the low-dispersal cell mixture can be found at https://doi.org/10.5281/zenodo.8170426.

## ADDITIONAL FILES

The following material is available online.

### Supplemental Material

**Figure S1 (mSystems00425-23-s0001.pdf).** Measures of the cell behaviors that drive aggregation.
**Figure S2 (mSystems00425-23-s0002.pdf).** Reversal bias.
**Figure S3 (mSystems00425-23-s0003.pdf).** Reversal bias.
**Figure S4 (mSystems00425-23-s0004.pdf).** Reversal bias.
**Figure S5 (mSystems00425-23-s0005.pdf).** Aggregate dispersal.
**Figure S6 (mSystems00425-23-s0006.pdf).** Aggregate count.
**Figure S7 (mSystems00425-23-s0007.pdf).** Aggregate count.
**Figure S8 (mSystems00425-23-s0008.pdf).** Aggregate count.
**Movie S1 (mSystems00425-23-s0009.mp4).** High-dispersal strain DK10547 from 10-24 hours. Corresponds to HD data set 1.
**Movie S2 (mSystems00425-23-s0010.mp4).** Low-dispersal strain S4 from 12-24 hours. Corresponds to LD data set 2.

### Open Peer Review

**PEER REVIEW HISTORY (review-history.pdf).** An accounting of the reviewer comments and feedback.

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
