## [Reviewer comments · mSystems]

Cell behaviors underlying *Myxococcus xanthus* aggregate coarsening

Patrick Murphy, Jessica Comstock, Trosporsha Khan, Jianguo Zhang, Roy Welch, and Oleg Igoshin

Corresponding Author(s): Oleg Igoshin, Rice University

Review Timeline:

Submission Date:	May 8, 2023
Editorial Decision:	June 14, 2023
Revision Received:	July 24, 2023
Accepted:	July 27, 2023

Editor: Alejandra Rodríguez-Verdugo

Reviewer(s): The reviewers have opted to remain anonymous.

Transaction Report:

DOI: <https://doi.org/10.1128/msystems.00425-23>

June 14, 2023

Prof. Oleg A Igoshin
Rice University
Bioengineering
MS142
P.O. Box 1892
Houston, Texas 77005

Re: mSystems00425-23 (Cell behaviors underlying *Myxococcus xanthus* aggregate coarsening)

Dear Prof. Oleg A Igoshin:

Thank you for submitting your manuscript to mSystems. We have completed our review and I am pleased to inform you that, in principle, we expect to accept it for publication in mSystems. However, acceptance will not be final until you have adequately addressed the reviewer comments.

The two reviewers and I agree that this study addresses important questions on microbial collective behaviors and will advance our understanding of Myxobacteria biology. The two reviewers have made excellent suggestions for improvement that should be addressed before we can accept your work for publication. Reviewer #1 provided some suggestions to strengthen the study, e.g., doing experiments in slow-flow chambers to support some of the hypothesized mechanisms. If this is not possible, please discuss the value of these experiments for follow-up studies. I am looking forward to receiving a revised manuscript.

Preparing Revision Guidelines

Please return the manuscript within 60 days; if you cannot complete the modification within this time period, please contact me. If you do not wish to modify the manuscript and prefer to submit it to another journal, please notify me of your decision immediately so that the manuscript may be formally withdrawn from consideration by mSystems.

Sincerely,

Alejandra Rodríguez-Verdugo

Editor, mSystems

Journals Department
Reviewer comments:

Reviewer #1 (Comments for the Author):

This paper examines the collective behavior of starved colonies of *Myxococcus xanthus*, which is a model organism for studying fundamental features of cell-signaling and cell differentiation in a bacterial community. While a great deal of research has already been done on the precursors to fruiting body formation in this system, there remains a great deal that we still do not understand. This paper focuses on stabilization of cell aggregates, in which case some aggregates remain and eventually develop into fruiting bodies while others disperse. By quantitatively examining motility parameters of individual cells near stable and unstable aggregates, the authors identify two main changes to the cell behavior that drive aggregate stability, cell reversal bias and the propensity for entering and staying in the non-persistent state, and that these parameters are influenced by the area of the cell aggregate. Using an agent-based model, the authors show that (a) using a model where motility parameters are altered by proximity to an aggregate and the size of the aggregate that they can reproduce much of the experimentally-determined statistics on aggregate stability, size, etc, and (b) cell reversal bias is the dominant parameter that determines aggregate stability. Overall, this is a well-written paper that further advances our understanding of Myxobacteria.

I do not have any major criticism about the results reported here, but do offer a few comments for the authors to consider.

1. The choice of focusing on one experimental dataset to determine the motility alterations and to compare the initial modeling results to is ok, especially since the authors then use those findings to compare to a couple other datasets. However, is there a way to create an average picture that summarizes the general behavior over multiple experiments? See also comment 3 below.
2. The results seem to point strongly toward a diffusible chemoattractant as the mechanism by which the cells would sense proximity to and size of cell aggregates. Is there further evidence that would support this? If this is the case, then carrying out the experiments in a slow flow chamber should alter the aggregate dynamics. Is this doable and/or has it been done? If the experiment is easy enough to do, the authors may want to consider doing the experiment.
3. I found the result described on lines 296-300 to be interesting and informative, yet the authors don't delve into its implications. Specifically, the authors find that in examining different experiments that the size of unstable aggregates is based on relative size of the aggregate to the average size across the system, as opposed to being dependent on absolute size. This would suggest that the internal cell signal response is adaptive (that the response is based on differences from a baseline signal, where the baseline adjusts in response to the total amount of signal), similar to what has been observed in *Dictyostelium* chemotaxis (and probably others). This should be informative about what possible biochemical mechanisms for the signaling pathway are.

Reviewer #2 (Comments for the Author):

This manuscript presents a quantification of reversal bias of *M. xanthus* and its role in the formation and dispersal of aggregates. Specific comments are below.

1. Fig. 1: One representative image is shown. Where is the statistical strength? Is it always 11.5 h? What is the spread? Could it be 10, 12, or 13 h? Fig.1 is cited during the introduction section. In case the 11.5 h data is from some other work, that work should be cited.
2. Minor: page 6 line 122: there seems to be a typing error. Remove 'Additional divisions were applied'
3. Page 7, line 134: What is a *Myxococcus* mixture? Is it a different strain. Are these different strains in different ratios. Please explain it in the text.
4. Page 7, Line 141 onwards: The calculation of reversal bias only considers a binary scenario. Do cells only move towards or away from the aggregates? As an example do some cells move orthogonal or at a 45 degree angle from the direction of the chemotaxis gradient? How do the authors decide if such cells should be binned as $t(\text{toward})$ or $t(\text{away})$?
5. Page 8, line 160: What type of simulation? Is it the ABM mentioned in methods?
6. Page 10, line 195: it would be nice if the reader is reminded that the coarsening phase occurs around 11.5 h.

7. Page 10, line 196: cite the figure (3A?)

Dear Editor,

We are excited to resubmit the revised manuscript for publication in mSystems. We thank the reviewers for their time and thoughtful comments. Our revised and improved manuscript addresses these comments. We include a version of the manuscript with major changes highlighted in yellow for your convenience. Our specific responses are below in blue font. The corresponding changes to the manuscript are in red.

Response to Reviewer #1:

This paper examines the collective behavior of starved colonies of *Myxococcus xanthus*, which is a model organism for studying fundamental features of cell-signaling and cell differentiation in a bacterial community. While a great deal of research has already been done on the precursors to fruiting body formation in this system, there remains a great deal that we still do not understand. This paper focuses on stabilization of cell aggregates, in which case some aggregates remain and eventually develop into fruiting bodies while others disperse. By quantitatively examining motility parameters of individual cells near stable and unstable aggregates, the authors identify two main changes to the cell behavior that drive aggregate stability, cell reversal bias and the propensity for entering and staying in the non-persistent state, and that these parameters are influenced by the area of the cell aggregate. Using an agent-based model, the authors show that (a) using a model where motility parameters are altered by proximity to an aggregate and the size of the aggregate that they can reproduce much of the experimentally-determined statistics on aggregate stability, size, etc, and (b) cell reversal bias is the dominant parameter that determines aggregate stability. Overall, this is a well-written paper that further advances our understanding of Myxobacteria.

I do not have any major criticism about the results reported here, but do offer a few comments for the authors to consider.

1. The choice of focusing on one experimental dataset to determine the motility alterations and to compare the initial modeling results to is ok, especially since the authors then use those findings to compare to a couple other datasets. However, is there a way to create an average picture that summarizes the general behavior over multiple experiments? See also comment 3 below.

Thank you. We experimented with both combining cell behaviors for the data analysis and for the agent-based simulations. The issues we encountered were that the sizes of unstable aggregates, the start of the coarsening phase, and the time when the mean reversal bias drops to its lowest value all varied across experimental replicates. We had some success with scaling time and mean aggregate area to combine the data sets, especially for the data analysis. However, the simulations running on combined data only had about 30% of aggregates dispersing, rather than the ~50% observed in experiments. We suspect scaling and combining data reduced or eliminated certain correlations responsible for aggregate formation and dispersal. For example, we cannot linearly scale time so that the start of initial aggregation, the start of the coarsening phase, and the peak drop in the reversal bias all match. If certain cell behaviors or correlations are most prominent in a shorter range of time (1-2 hours), combining data sets will reduce their strength. Therefore focusing on individual data sets (1) avoids

introducing this issue to our analysis and demonstrates (2) that the model can match the observed trends in each dataset based on sampling from cell behaviors from that dataset.

We have included examples of the reversal bias near aggregates of different stability with the three wild-type data sets combined. The reversal bias is plotted both in time (Fig. S4a in the paper and at the end of this response) and in space (Fig. S4a). Using stability avoids the need to scale aggregate areas to combine the data faithfully, but still illustrates that some of the features of the combined reversal bias are smoothed out or weakened in strength compared to the originals.

2. The results seem to point strongly toward a diffusible chemoattractant as the mechanism by which the cells would sense proximity to and size of cell aggregates. Is there further evidence that would support this? If this is the case, then carrying out the experiments in a slow flow chamber should alter the aggregate dynamics. Is this doable and/or has it been done? If the experiment is easy enough to do, the authors may want to consider doing the experiment.

Thank you for suggesting this. First of all, we would like to emphasize that experiments presented here report aggregates on top of agar without any fluid (other than that the cells may have pulled out), and therefore diffusible communication must occur through agar and/or the biofilm matrix. There are also standard protocols for inducing aggregate formation in myxobacteria in a submerged culture (*"Dynamics of Fruiting Body Morphogenesis"* Kaiser and Welch 2003), where cells are placed on agar with a layer of liquid that provides a dwindling supply of nutrients covering the cells. Aggregation is possible in these cultures, although the degree to which the aggregates disperse in a submerged culture is not qualified.

The idea of a flow chamber is very interesting. However, there are also some aspects to consider regarding the expected results of experiments in a slow-flow chamber. The presence of aggregation suggests it is unlikely a slow-flow chamber would significantly weaken or eliminate a diffusible signal if one is responsible for maintaining aggregate cohesion. Perhaps strengthening the flow or adding chemicals to test the response from cells would work. The closest previous study we are aware of along these lines is *"Growth of Myxococcus xanthus in Continuous-Flow-Cell Bioreactors as a Method for Studying Development"* Smaldone et al. 2014. This study looked at the development of *M. xanthus* communities in flow cells, which they used to test the response of the communities to environmental stimuli such as nutrients. Flow cells are rather finicky to work with and take a long time to develop protocols for. These considerations make flow cells infeasible to pursue for this particular study. Lastly, pursuing chemotaxis would take away from the points we wanted to make in this paper (namely reversal bias driving aggregate dispersal) simply due to it being a research topic with a scope easily spanning multiple papers and experiments.

A cohesive approach to determining the biochemical source of aggregate dispersal would necessitate different approaches such as chemical (fractionating cells and testing the response of cells to different molecular subsets) and genetic (finding a mutant that doesn't develop or doesn't coarsen). We anticipate pursuing at least some of these approaches in future work.

We have added the following to the discussion regarding ideas for future experiments: "We propose that this approach to identifying the biological source of the reversal bias could be supplemented by additional mutant screening, fractionating cells to test the response of

aggregates to different molecular subsets, and testing how aggregation dynamics vary in different chemical and physical environments using a slow-flow chamber.”

3. I found the result described on lines 296-300 to be interesting and informative, yet the authors don't delve into its implications. Specifically, the authors find that in examining different experiments that the size of unstable aggregates is based on relative size of the aggregate to the average size across the system, as opposed to being dependent on absolute size. This would suggest that the internal cell signal response is adaptive (that the response is based on differences from a baseline signal, where the baseline adjusts in response to the total amount of signal), similar to what has been observed in *Dictyostelium* chemotaxis (and probably others). This should be informative about what possible biochemical mechanisms for the signaling pathway are.

Thank you for pointing this out. We are considering possible biochemical mechanisms with an eye on future experiments. We have added the following in the discussion exploring the implications of the size threshold for aggregate dispersal changing with the mean size of all aggregates: “... the average size of unstable aggregates is based on their relative size to the average size of all aggregates. Therefore, the signaling pathway would need to be capable of population-level sensing since the size threshold for unstable aggregates scales with the size of all aggregates (Fig. 6A). Therefore, the reversal bias would result from local differences from a baseline signal across the whole population.”

Response to Reviewer #2:

This manuscript presents a quantification of reversal bias of *M. xanthus* and its role in the formation and dispersal of aggregates. Specific comments are below.

1. Fig. 1: One representative image is shown. Where is the statistical strength? Is it always 11.5 h? What is the spread? Could it be 10, 12, or 13 h? Fig.1 is cited during the introduction section. In case the 11.5 h data is from some other work, that work should be cited.

Thank you for bringing this up. Fig. 1 is just intended to illustrate the initial aggregation and coarsening phases. The 11.5 hour mark in Fig. 1 is for the start of the coarsening phase in that particular experiment (referred to as HD data set 1), which was performed for this paper. The exact start of the coarsening phase varies from replicate to replicate, but for us was typically 9-12 hours after plating cells on agar (vertical lines in Fig. 6B). Other studies have reported values of 11 hours after plating on average over 20 experiments (“*Describing Myxococcus xanthus Aggregation Using Ostwald Ripening Equations for Thin Liquid Films*” Bahar *et al.* 2014) and 13.5 hours after plating (“*Statistical image analysis reveals features affecting fates of Myxococcus xanthus developmental aggregates*” Xie *et al.* 2011). It is worth pointing out that coarsening happens concurrently across the field of view. Lastly, we want to point out the timing of the start of the coarsening phase is only used in a few parts of the paper. These are the reversal bias plotted in Fig. S6f and in simulations of the agent-based model when determining cellular behaviors based on the current phase as outlined in the methods section.

We have added a sentence with references to clarify when the coarsening phase typically starts: “The start of the coarsening phase varies but is typically 10-15 hours after plating cells on agar.” Additionally, we have added “The beginning of the coarsening phase varies by experiment, and was calculated here as outlined in Materials and Methods.” to the caption of Fig. 1 to point readers to how we determined the start of the coarsening phase.

2. Minor: page 6 line122: there seems to be a typing error. Remove 'Additional divisions were applied'

Thank you for pointing this out. 'Additional divisions were applied' has been removed.

3. Page 7, line 134: What is a Myxococcus mixture? Is it a different strain. Are these different strains in different ratios. Please explain it in the text.

Thank you for pointing this out. We meant to state there that our mixture of fluorescent strains (1:800 ratio of tdTomato:GFP labeled cells from strains LS3908 and DK10547 respectively as covered in the methods section) differed from the ones used in the previous cited study on cellular behaviors during aggregation. We have clarified this in the text.

4. Page 7, Line 141 onwards: The calculation of reversal bias only considers a binary scenario. Do cells only move towards or away from the aggregates? As an example do some cells move orthogonal or at a 45 degree angle from the direction of the chemotaxis gradient? How do the authors decide if such cells should be binned as t(toward) or t(away)?

Thank you for your question. We observed cells moving at nearly all angles relative to aggregates. In this manuscript we binned cells into categories of towards and away using a threshold of 90 degrees in magnitude from the direction pointing towards the center of the aggregate. Angles of -90 to 90 degrees were considered as “towards”, while angles greater in magnitude were considered as “away”. There were cells with orientations nearly orthogonal to the nearest aggregate, but the majority of these were located at the aggregate boundary and did not affect measurements of bias farther away from the boundary.

We have included an example of our data analysis for the reversal bias (updated Fig. S2a & S2b in the paper and at the end of this response) removing angles between 45 and 135 degrees in magnitude relative to the aggregate. There are some small changes in these figures, but nothing significant has changed.

Lastly, we want to note the simulations did not use a binary classification for cells' relative orientations to aggregates. Relative orientations for agent cells were not binned into categories to determine simulated cell behavior. Instead, behaviors were sampled from experimental data based on a nearest-neighbors search using the exact relative orientation of each simulated cell.

5. Page 8, line 160: What type of simulation? Is it the ABM mentioned in methods?

Thank you for pointing this out. The simulations were using the agent-based model outlined in the methods and references (Cotter et al. 2017). We have made this explicit in the text.

6. Page 10, line 195: it would be nice if the reader is reminded that the coarsening phase occurs around 11.5 h.

Thank you. We have included a reminder of the start of the coarsening phase.

7. Page 10, line 196: cite the figure (3A?)

Thank you. We have included a citation there to figure 3A.

Fig. S4. A) Combined reversal bias from all three HD data sets for cells near large (blue) and small (red) aggregates versus time. Data sets are aligned in time at hour 0 based on the start of the coarsening phase. B) Combined reversal bias from all three HD data sets versus the distance from the nearest aggregate's boundary (set to be 0) for cells near stable (blue) and unstable (red) aggregates in the experiment. Data sets are again aligned in time based on the start of the coarsening phase. The shaded regions in all panels denote 95% confidence intervals for the mean, and the vertical red lines mark the start of the coarsening phase. Figure labels are corrected from C and D

Fig. S2A&B. A) Plot of the reversal bias for cells near large (blue) and small (red) aggregates versus time with cells oriented 45-135 degrees in magnitude from the nearest aggregate removed. B) Reversal bias versus the distance from the nearest aggregate's boundary (set to be 0) for cells near stable (blue) and unstable (red) aggregates in the experiment. Like in A), cells oriented 45-135 degrees in magnitude from the nearest aggregate have been removed.

July 27, 2023

Prof. Oleg A Igoshin
Rice University
Bioengineering
MS142
P.O. Box 1892
Houston, Texas 77005

Re: mSystems00425-23R1 (Cell behaviors underlying Myxococcus xanthus aggregate coarsening)

Dear Prof. Oleg A Igoshin:

Thank you for your revisions and for carefully responding to the reviewers' comments. These changes have strengthened the manuscript. The manuscript is now ready for publication! Thank you for the privilege of reviewing your work.

Your manuscript has been accepted, and I am forwarding it to the ASM Journals Department for publication. For your reference, ASM Journals' address is given below. Before it can be scheduled for publication, your manuscript will be checked by the mSystems production staff to make sure that all elements meet the technical requirements for publication. They will contact you if anything needs to be revised before copyediting and production can begin. Otherwise, you will be notified when your proofs are ready to be viewed.

If you would like to submit a potential Featured Image, please email a file and a short legend to msystems@asmusa.org. Please note that we can only consider images that (i) the authors created or own and (ii) have not been previously published. By submitting, you agree that the image can be used under the same terms as the published article. File requirements: square dimensions (4" x 4"), 300 dpi resolution, RGB colorspace, TIF file format.

We recognize that the video files can become quite large, and so to avoid quality loss ASM suggests sending the video file via <https://www.wetransfer.com/>. When you have a final version of the video and the still ready to share, please send it to mSystems staff at msystems@asmusa.org.

Sincerely,

Alejandra Rodríguez-Verdugo
Editor, mSystems

Journals Department
E-mail: mSystems@asmusa.org